# Cardiorespiratory Fitness and Muscular Strength Moderates the Relationship between *FNDC5* Polymorphism and Adiposity in Children and Adolescents

**DOI:** 10.3390/ijerph18189797

**Published:** 2021-09-17

**Authors:** Pâmela Ferreira Todendi, Caroline Brand, João Francisco de Castro Silveira, Ryan Donald Burns, J. Alfredo Martínez, Marilu Fiegenbaum, Anelise Reis Gaya, Jane Dagmar Pollo Renner, Cézane Priscila Reuter, Andréia Rosane de Moura Valim

**Affiliations:** 1Graduate Program in Pathology, Federal University of Health Sciences of Porto Alegre—UFCSPA, Sarmento Leite St, 245, Centro Histórico, Porto Alegre 90050-170, Brazil; pamelaferreiratodendi@gmail.com; 2Graduate Program in Health Promotion, University of Santa Cruz do Sul, Independência Av. 2293-Universitário, Santa Cruz do Sul 96815-900, Brazil; carolbrand@hotmail.com.br (C.B.); joaofrancisco@hotmail.com (J.F.d.C.S.); janerenner@unisc.br (J.D.P.R.); avalim@unisc.br (A.R.d.M.V.); 3Department of Health, Kinesiology, and Recreation, University of Utah, 201 Presidents’ Cir, Salt Lake City, UT 84112, USA; Ryan.D.Burns@utah.edu; 4Department of Nutrition and Food Sciences, Physiology and Toxicology, University of Navarra, 28083 Madrid, Spain; jalfmtz@unav.es; 5Department of Basic Health Sciences, Federal University of Health Sciences of Porto Alegre—UFCSPA, Sarmento Leite St, 245, Centro Histórico, Porto Alegre 90690-200, Brazil; mariluf@ufcspa.edu.br; 6Graduate Program in Human Movement Sciences, Federal University of Rio Grande do Sul, Porto Alegre 90690-200, Brazil; anegaya@gmail.com

**Keywords:** genetics, physical fitness, obesity, children, adolescents

## Abstract

The human locus *FNDC5* rs16835198 contributes positively to anthropometric phenotypes in children and adolescents. However, the role of specific components of physical fitness in this relationship is not known. The present study aimed to verify the moderator role of cardiorespiratory fitness (CRF) and muscular strength in the relationship between rs16835198 polymorphism *FNDC5* and adiposity in children and adolescents. This cross-sectional study was carried out by genotyping the rs16835198 *FNDC5* polymorphism in 1701 children and adolescents (mean age 11.73 ± 2.75 years). Obesity was assessed using waist circumference and body mass index (BMI) z-scores. To evaluate CRF and muscular strength, the 6 min run/walk test and lower limb strength (LLS) were used. Linear regression models were applied, and all analyses were adjusted for age, sex, skin color, living area, and school type. A significant interaction term for CRF (*p* = 0.038) and LLS (*p* = 0.040) × rs16835198 *FNDC5* with WC was identified. Regarding BMI, a significant interaction term for CRF (*p* = 0.007) and LLS (*p* = 0.044) × rs16835198 *FNDC5* was observed. Moreover, medium and high CRF and LLS levels protected against higher WC and BMI. In conclusion, adiposity levels of children and adolescents with a genetic predisposition to obesity might be modified by improving CRF and muscular strength.

## 1. Introduction

Prevalence of obesity among children and adolescents is high, and rates have increased over the past decade [1]. Globalization has led to an increase in the so-called “Western” food pattern characterized by the consumption of foods with large amounts of refined carbohydrates, sugars, salt, and saturated fats [2]. This phenomenon created a “nutritional transition”, and problems such as overweight, obesity, and chronic diseases have been noted in childhood and adolescence [1].

In order to mitigate obesity prevalence, physical activity practices have been vital to reduce cardiometabolic risk among young people [3]. The benefits of exercise are mediated by the metabolic and molecular remodeling of the skeletal muscle [4] and also by the release of cytokines from the muscle, called myokines [5]. Among the biologically active substances secreted by myocytes, irisin, which is encoded by the *FNDC5* gene [6], is a pleiotropic hormone that mediates the beneficial effects of physical activity, such as increased energy expenditure and fat oxidation. Human studies have shown that exercise significantly increases irisin concentrations [7,8]. Therefore, it is likely that the health benefits of exercise are, at least in part, due to the expression of *FNDC5* induced by skeletal muscle exercise and the consequent increase in circulating irisin concentration [6].

Another important effect of physical exercise is the increase in cardiorespiratory fitness (CRF) and muscular strength, the most studied components of physical fitness, which are considered important health indicators in all ages [9,10]. CRF and muscular strength are inversely associated with adiposity and exert a protective role against the development of cardiometabolic risk factors in children and adolescents [11,12]. Since physical activity and exercise improves CRF, it is plausible that the effects of CRF are mediated by irisin [8].

This association suggests that *FNDC5* is beneficial for the treatment of obesity, diabetes, and a variety of pathological conditions characterized by an imbalance in energy expenditure [13]. It is observed that the human locus *FNDC5* rs16835198 contributed positively to anthropometric phenotypes in a previous Brazilian study [14]. However, the role of specific components of physical fitness in the relationship between *FNDC5* polymorphism and adiposity is not known. In this sense, the aim of this study was to verify the moderator role of CRF and muscular strength in the relationship between rs16835198 polymorphism *FNDC5* and adiposity in children and adolescents.

## 2. Materials and Methods

### 2.1. Study Design and Sample

All children and adolescents from 25 randomly selected schools in Santa Cruz do Sul were recruited between 2014 and 2015. The students who chose to participate were the initial sample (*n* = 2200). From the initial sample, 1701 had the rs16835198 Irisin/*FNDC5* polymorphism genotyped. This cross-sectional research was composed of children and adolescents between 7 and 17 years old belonging to the Schoolchildren’s Health Study—Phase III (Santa Cruz do Sul, Rio Grande do Sul, Brazil). The Schoolchildren’s Health Study is a lifestyle education program supported by a multidisciplinary team of nutritionists, nurses, pharmacists, physiotherapists, and physical educators from the University of Santa Cruz do Sul (UNISC). This study meets the ethical standards established by the Declaration of Helsinki and was approved by the Research Ethics Committee of the UNISC (number 714 216/14).

A written consent, signed in two copies, was obtained from parents or guardians prior to the beginning of the study. Students and/or parents were interviewed face-to-face, and a questionnaire was completed. The questionnaire comprised questions regarding students’ demographic data (i.e., age, ethnicity, sex, living area, school network, etc.) and lifestyle habits (i.e., practice of physical activity, hours of sleep per night, etc.). Brazil is a country with great miscegenation [15], and for this reason participants’ ethnicity determination was conducted according to Parra et al. [16], based on an evaluation of the following phenotypic characteristics: skin color in the medial part of the arm, color and texture of hair, and the shape of the nose and lips.

### 2.2. Measurements

Weight and height were measured on the anthropometric scale with a coupled stadiometer (Filizola^®^, São Paulo, Brazil). BMI was then calculated by dividing body mass (in kilograms) by height (in square meters). Waist circumference (WC) was measured using a flexible non-elastic tape with a resolution of 1 mm (Cardiomed^®^, Curitiba, Brazil), parallel to the ground and placed midway between the iliac crest and the last rib.

The described procedures of CRF and muscular strength evaluation followed the “Projeto Esporte Brasil” standard [17]. CRF was evaluated by a six-minute walk/run test. Children and adolescents should accomplish the greatest number of turns, running or walking, in a sports court with the perimeter marked with cones and the floor with indications of meters. The number of laps successfully completed, plus the additional distance achieved for the ones unable to complete a full lap at the termination of the test, was calculated. The result of the test was obtained by multiplying the number of laps by meters covered. In the present study, lower limb strength (LLS) was considered as a measure of muscular strength. LLS tests were carried out with a measuring tape fixed to the floor. The start line was marked using chalk, with the zero point of the measuring tape being placed in line with the chalk mark. The children/adolescents were positioned immediately behind the starting line, with feet parallel and knees partially flexed. At the signal, participants were instructed to jump as far as possible with both feet at the same time.

### 2.3. Genotyping

The DNA was extracted using the salting out method [18]. Later, it was quantified using the NanoDrop 2000c spectrophotometer unit (Thermo Scientific, Wilmington, NC, USA). The genotyping of polymorphism *FNDC5* rs16835198 was performed using Taq-man™ allelic discrimination assay (Applied Biosystems, Foster City, CA, USA) in StepOne Plus^®^ equipment, according to the manufacturer’s instructions. TaqMan™ assay C__34204885_10 (rs16835198) and Master Mix PCR Universal were purchased from Applied Biosystems.

### 2.4. Statistical Analyses

Descriptive data are presented as means, standard deviation, and frequency. Independent two-tailed *t*-tests and chi-square tests were used to examine sex differences for continuous and categorical data, respectively. The effect size (Cohen’s *d*) was also calculated to determine differences in means between sexes, considering its variability.

Moderation analyses were conducted using linear regression models via the PROCESS macro for the Statistical Package for Social Sciences (SPSS) version 24.0 (IBM Corp, Armonk, NY, USA). The following models were tested: (1) Association between CRF and *FNDC5* rs16835198 and interaction (CRF × *FNDC5* rs16835198) with adiposity parameters. (2) Associations between LLS and *FNDC5* rs16835198 and interaction (LLS × *FNDC5* rs16835198) with adiposity parameters. TT allele was considered the risk category for all analyses. To establish the moderation point, meaning the point of CRF and LLS in which there was no association between *FNDC5* rs16835198 and adiposity parameters, the Johnson–Newman technique was considered, and CRF and muscular strength were classified according to tertiles (low, medium, and high). All analyses were adjusted for age, sex, skin color, living area, and school network. The probability value *p* < 0.05 was considered to be significant for all analyses.

## 3. Results

Table 1 presents the characteristics of the participants by sex. Boys presented higher mean values of height, waist circumference, cardiorespiratory fitness, and lower limb strength, while girls showed higher mean values of BMI.

The moderator role of CRF and muscular strength in the relationship between rs16835198 polymorphism (*FNDC5*) and adiposity parameters is presented in Table 2. Results indicated that CRF was inversely associated with WC, and children and adolescents with the TT allele were more likely to present higher WC. A significant interaction term was found for CRF × *FNDC5* rs16835198 with WC. Regarding muscular strength, LLS was inversely associated with WC, and children and adolescents with the TT allele were more likely to present higher WC. Additionally, a significant interaction term was found for LLS × *FNDC5* rs16835198 with WC. Similar findings were observed for BMI in both CRF and LLS models, with a significant interaction term for LLS and CRF × *FNDC5* rs16835198 with BMI. Thus, CRF and muscular strength are moderators in the relationship between *FNDC5* rs16835198 with WC and BMI.

Considering the interaction between CRF × rs16835198 *FNDC5* with WC and BMI, we aimed to establish from which point of CRF there was protection against high values of these variables, considering the rs16835198 *FNDC5* (Figure 1 and Figure 2). Children and adolescents with the TT allele were more likely to present high WC, compared to the ones with the GT + GG allele. This association was observed only for the ones classified with low CRF (710.00 m) and low muscular strength (1.03 m). For children and adolescents with medium and high CRF and LLS, there was no longer an association between rs16835198 *FNDC5* and WC. Thus, children and adolescents that accomplished more than 836.89 m and 1.26 m in the CRF and LLS tests, respectively, were protected against WC, as there was no longer association with rs16835198 *FNDC5* (Figure 1).

Similar findings were observed considering BMI as the dependent variable (Figure 2). The association between rs16835198 *FNDC5* and BMI was only observed for children and adolescents with low CRF. The moderation point indicated that children who accomplished between 850 m and 1403 m in the CRF test were protected against high BMI, as there was no longer association with rs16835198 *FNDC5*. In addition, for the ones that accomplished more than 1403 m, the association between rs16835198 *FNDC5* became negative, indicating that the children and adolescents with the TT allele were less likely to present high BMI, compared to the ones with the GT + GG allele. Regarding LLS, results showed an association between rs16835198 *FNDC5* and BMI in low and medium values of the LLS test, indicating that the children and adolescents that accomplished more than 1.28 m were protected against high BMI. Therefore, appropriate levels of CRF and muscular strength are important factors against the deleterious effect of the TT allele on the increasing WC and BMI.

## 4. Discussion

The findings of this study indicate that CRF and muscular strength are moderators in the relationship between rs16835198 polymorphism *FNDC5* and adiposity in children and adolescents. Furthermore, medium and high CRF and LLS levels were not associated with WC and BMI when considering the influence of *FNDC5* rs16835198. For WC, this was observed when children achieved more than 836.89 m and 1.26 m in the CRF and LLS, respectively; whereas for BMI they should achieve more than 850 m (CRF) and 1.28 m (LLS). These findings have implications in terms of preventing high adiposity levels and consequent cardiometabolic disorders, especially when considering the genetic predisposition to obesity. To the best of our knowledge, this is the first study to assess the moderating role of CRF and muscular strength in the relationship between genetic susceptibility to the *FNDC5* gene and obesity in children and adolescents.

Irisin plays an important role in fat metabolism, and the secretion of this myokine is induced by physical exercise [19], suggesting that irisin mediates health benefits via “browning” of adipose tissue. Irisin is, therefore, associated with muscle mass, strength, and metabolism. However, the role of the expressed form of irisin, concerning physiological functions and circulating levels, remains controversial in humans [20]. Thus, the genetic evaluation of *FNDC5* can be useful to identify patterns of metabolic health and disease [21] with promising expectation in the treatment of obesity [20].

High dietary energy intake and low energy expenditure through physical activities are the main responsible factors for excess development of adiposity [22]. Obesity-related behaviors such as food consumption and physical activity seem to track from childhood to adulthood [23]. Therefore, youth populations need to maintain healthy behavioral habits from an early age, as it is well established that adiposity during childhood may be a risk factor for the development of adult cardiometabolic risk factors [24]. Our findings support the importance of CRF and muscular strength on attenuating adiposity development in children and adolescents with a genetic predisposition to obesity.

Other findings also demonstrate that healthier CRF levels seems to attenuate the adverse outcomes of excess adiposity on overall health [25]. There is evidence indicating that improving childhood CRF may improve overall health by reducing adiposity [26]. Moreover, there is also evidence demonstrating that muscular fitness improvements during childhood and adolescence might promote healthier adiposity levels [27]. However, caution should be exerted when interpreting the muscular fitness influence on adiposity, as this domain can be assessed with a variety of assessments (e.g., strength, power, or endurance). Nevertheless, our study reports the moderating role of LLS as a measure of muscular strength in the relationship between rs16835198 polymorphism *FNDC5* and adiposity in children and adolescents.

Children and adolescents are physiologically adaptable to aerobic and resistance exercises [28,29]. Based on this knowledge, our findings are particularly remarkable for targeting overall health, especially for those with a genetic predisposition to obesity. Thus, in order to increase CRF and muscular strength, and consequently reduce adiposity levels and improve metabolic health [27,30], children and adolescents should follow the recommendations of 60 min of moderate to vigorous physical activity every day, along with activities that maintain or improve muscular strength at least three times a week [31]. Moreover, it is worth highlighting that these findings bring important and novel contributions to the existing literature by determining the level of CRF and LLS that children and adolescents must achieve to attenuate the association between rs16835198 polymorphism *FNDC5* and adiposity.

However, it must be considered that the prevention and treatment of childhood obesity is a complex task. Additionally, it is known that this disease is of polygenic origin, and other genes are also associated with it, including *MCR4*, *LEP*, *FTO,* and *TMEM18* [14,32]. Knowing that increasing levels of physical activity is an important part of the process, we should mention mechanisms for promoting a more active lifestyle. In this sense, interventions focused on the family and at school have shown an important role in childhood adiposity, as the adoption of sedentary habits occurs primarily in the interpersonal context, such as at home and at school [33].

Our study has some worthwhile strengths. The presentation of a randomly selected sample of children and adolescents from a municipality in southern Brazil was a strength. As outlined earlier, this is the first study assessing the moderating role of physical fitness in the relationship between genetic susceptibility and youth obesity. In addition, our study provided a point from which CRF and muscular strength attenuates the excess of adiposity, when considering genetic susceptibility, allowing to more accurately indicate the level of these variables needed for a beneficial effect. However, some limitations should be noted. First, the cross-sectional approach does not allow us to establish causal relationships; therefore, the interpretation of the relationships must be analyzed with caution. We suggest the evaluation of the effect of physical fitness on the relationship with gene-obesity using longitudinal and experimental studies. Second, we evaluated only one polymorphism in the *FNDC5* gene; many other genes are known to be involved in the development of obesity. Third, we understand that there is sex specificity related to physical fitness levels and anthropometric characteristics, which we approached by adjusting the analysis for sex; however, future studies should consider testing the same relationship for boys and girls separately. Finally, the evaluation of *FNDC5* indicates the genetic susceptibility; however, we did not assess circulating levels of irisin.

## 5. Conclusions

CRF and muscular strength were moderators in the relationship between rs16835198 polymorphism *FNDC5* and adiposity in children and adolescents. This is the first study assessing the moderating role of physical fitness in the relationship between genetic susceptibility and youth obesity. Based on the current findings, adiposity levels in children and adolescents with a genetic predisposition to obesity might be modified by improving CRF and muscular strength performance, which should be considered when targeting children’s and adolescents’ overall health.

## Figures and Tables

**Figure 1 ijerph-18-09797-f001:**
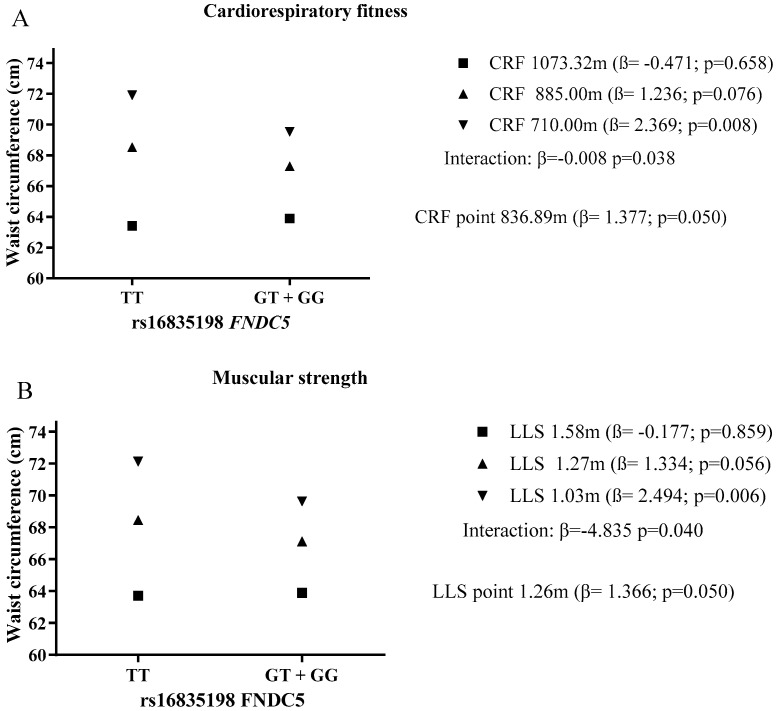
The role of cardiorespiratory fitness level (**A**) and muscular strength level (**B**) in the relationship between rs16835198 *FNDC5* and waist circumference. CRF: Cardiorespiratory fitness; LLS: Lower limb strength. All analyses were adjusted for age, sex, skin color, living area, and school network.

**Figure 2 ijerph-18-09797-f002:**
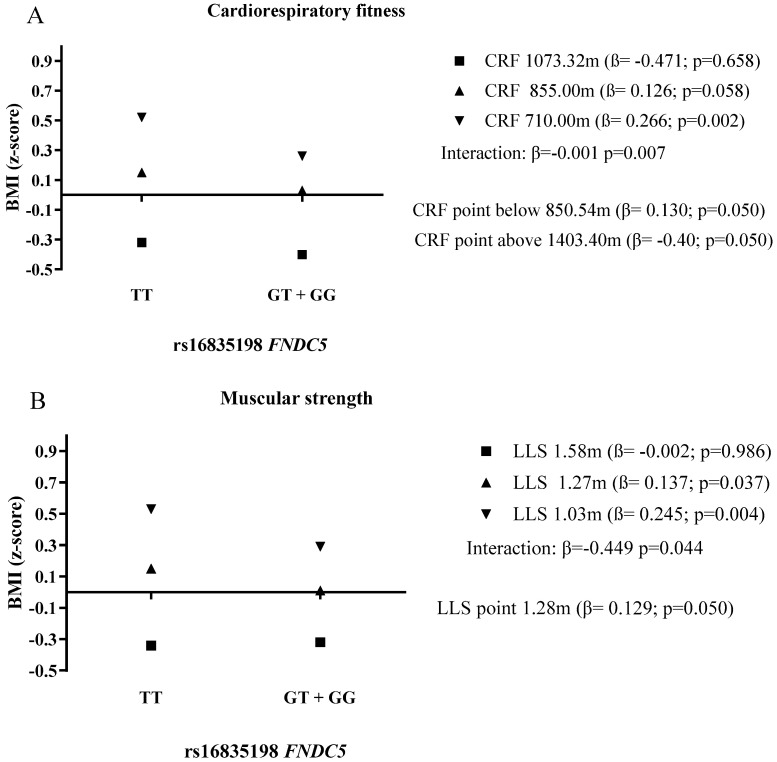
The role of cardiorespiratory fitness level (**A**) and muscular strength level (**B**) in the relationship between rs16835198 *FNDC5* and BMI. BMI: Body mass index; CRF: Cardiorespiratory fitness; LLS: Lower limb strength. All analyses were adjusted for age, sex, skin color, living area, and school network.

**Table 1 ijerph-18-09797-t001:** Baseline sample characteristics by sex.

	Mean (SD)			
Characteristics	Total (*n* = 1701)	Boys (*n* = 758)	Girls (*n* = 943)	Effect Size
Age (years)	11.73 (2.75)	11.58 (2.84)	11.85 (2.84)	0.09
Height (m)	1.50 (0.14)	1.51 (0.16)	1.50 (0.13) *	0.06
Weight (kg)	47.54 (15.95)	47.84 (17.34)	47.30 (14.74)	0.03
Body mass index (kg/m^2^)	20.43 (4.33)	20.19 (4.22)	20.62 (4.41) *	0.09
Waist circumference (cm)	67.03 (10.73)	67.78 (11.08)	66.42 (10.40) *	0.12
WHtR	0.44 (0.06)	0.44 (0.06)	0.44 (0.06)	0.00
Cardiorespiratory fitness (m)	885.07 (188.81)	971.09 (209.05)	814.94 (134.77) *	0.88
Lower limb strength (m)	1.30 (0.28)	1.41 (0.30)	1.21 (0.22) *	0.76
	*n* (%)			
rs16835198 *FNDC5*				
TT	220 (12.9)	86 (11.3)	134 (14.2)	-
GT + GG	1481 (87.1)	672 (88.7)	809 (85.8)	-
Living area				
Urban	1311 (77.1)	565 (74.5)	746 (79.2) **	-
Rural	389 (22.8)	193 (25.5)	196 (20.8)	-
School network				
Municipal	426 (25.0)	189 (24.9)	237 (25.1)	-
State	1239 (72.8)	551 (72.7)	688 (73.0)	-
Private	35 (20.1)	18 (2.4)	17 (1.8)	-

SD: Standard deviation; * independent *t*-test; ** Chi-square test for differences between boys and girls (*p* < 0.05).

**Table 2 ijerph-18-09797-t002:** Moderation of cardiorespiratory and muscular strength in the relationship between rs16835198 *FNDC5* and adiposity parameters.

KERRYPNX	Waist Circumference (cm)			
	β	CI (95%)	*p*	R^2^
Cardiorespiratory fitness				0.224
CRF	−0.016	−0.019; −0.013	<0.001	
rs16835198 *FNDC5*				
TT	7.918	1.409; 14.427	0.017	
GT + GG	1			
CRF × rs16835198 *FNDC5*	−0.008	−0.015; −0.001	0.038	
Muscular strength				0.218
LLS	−10.447	−12.538; −8.356	<0.001	
rs16835198 *FNDC5*				
TT	7.474	1.442; 13.507	0.015	
GT + GG	1			
LLS x rs16835198 *FNDC5*	−4.835	−9.445; −0.225	0.040	
	BMI (z-score)			
Cardiorespiratory fitness				0.201
CRF	−0.002	−0.002; −0.001	<0.001	
rs16835198 *FNDC5*				
TT	0.951	0.332; 1.570	0.003	
GT + GG	1			
CRF × rs16835198 *FNDC5*	−0.001	−0.002; 0.001	0.007	
Muscular strength				0.197
LLS	−1.147	−1.345; −0.949	<0.001	
rs16835198 *FNDC5*				
TT	0.707	0.136 1.277	0.015	
GT + GG	1			
LLS × rs16835198 *FNDC5*	−0.449	−0.885; −0.012	0.044	

CRF: Cardiorespiratory fitness; LLS: Lower limbs strength; BMI: Body mass index. All analyses were adjusted for age, sex, skin color, living area, and school type.

## Data Availability

The data presented in this study are available on reasonable request from the corresponding author.

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
