# Peer review of "Cardiorespiratory Fitness and Muscular Strength Moderates the Relationship between FNDC5 Polymorphism and Adiposity in Children and Adolescents"

_ijerph, 2021, doi:10.3390/ijerph18189797_

Round 1

Reviewer 1 Report

I have carefully read the entire article of the Authors. I really liked it. Synthetic Abstract, Introduction and Methods quite clear and unattainable. I would have a small observation: yes, we agree that a correlation is identified between CRF and fitness and the respective polymorphism, but ... what would be the possible future methods to treat or prevent obesity / excess fat deposition in children. I would like some more elaborate discussions. In addition, I would like to note if there have been similar activities. research in this regard. Thanks.

Author Response

Response: Thank you for your comment. We added this excerpt to the discussion, as we consider the role of parents and schools in the prevention and treatment of childhood obesity to be fundamental.

However, it must be considered that the prevention and treatment of childhood obesity is a complex task. Knowing that increasing levels of physical activity is an important part of the process, we should mention mechanisms for promoting a more active lifestyle. In this sense, interventions focused on the family and at school have shown an important role in childhood adiposity, as the adoption of sedentary habits occurs primarily in the interpersonal context, such as at home and at school (TOPHAM et al., 2021).

Reviewer 2 Report

To authors:

  1. According to UN, youth age is between 15-24. In this manuscript, the authors used the data from children aged between 7-17 years. This could be modified in the title.
  2. The interpretation that there is a direct relationship should be made cautiously. Since no experimental data is provided to examine the effect of physical fitness.
  3. Further, authors must comment on gender specificity with respect to this relationship.
  4. The authors focus only on one polymorphism in the FNDC5 gene, other genes may be involved in the development of obesity. The authors must discuss this in the discussion.
  5. Irisin levels were not measured therefore the evaluation of FNDC5 as a genetic susceptible factor should be thoroughly investigated.

Author Response

According to UN, youth age is between 15-24. In this manuscript, the authors used the data from children aged between 7-17 years. This could be modified in the title.

Response: Thank you for the observation and correction. The title was modified:

Cardiorespiratory fitness and muscular strength moderates the relationship between FNDC5 polymorphism and adiposity in children and adolescents.

The interpretation that there is a direct relationship should be made cautiously. Since no experimental data is provided to examine the effect of physical fitness.

Response: The reviewer is absolutely right, this aspect was corrected.

Further, authors must comment on gender specificity with respect to this relationship.

Response: We understand your concern and we have included this aspect as a limitation of the study.

The authors focus only on one polymorphism in the FNDC5 gene, other genes may be involved in the development of obesity. The authors must discuss this in the discussion.

Response: Thanks for the suggestion. The following sentence was inserted in the text: Additionally, it is known that this disease is of polygenic origin, and other genes are also associated with it, including MCR4, LEP, FTO and TMEM18 (RASKILIENE et al., 2021; TODENDI et al., 2018).

Irisin levels were not measured therefore the evaluation of FNDC5 as a genetic susceptible factor should be thoroughly investigated.

Response: Since circulations levels and physiological functions of irisin remains controversial in humans, we justified the use of genetic evaluation of FNDC5 in the discussion:

Irisin plays an important role in fat metabolism and the secretion of this myokine is induced by physical exercise [20], suggesting that irisin mediates health benefits via “browning” of adipose tissue. Irisin is, therefore, associated with muscle mass, strength and metabolism. However, the role of expressed form of irisin, concerning physiological functions and circulating levels, remains controversal in humans (LI et al., 2021). Thus, the genetic evaluation of FNDC5 can be useful to identify patterns of metabolic health and disease [21] with promising expectation in the treatment of obesity (LI et al., 2021).

Round 2

Reviewer 2 Report

The manuscript is improved. I have no further comments.